# Perceptions of electronic health records by ethnically diverse groups in mental health: A systematic literature review

Ammarah Ikram[1], Sahdia Parveen[2] and Eleftheria Vaportzis[1] 

[1]Department of Psychology, School of Law and Social Sciences, Faculty of Management, Sciences, and Engineering, University of Bradford, Bradford, UK and [2]Centre for Applied Dementia Studies, Faculty of Health and Social Care, University of Bradford, Bradford, UK

Electronic Health Records; Digital Health; Health Information Technology; Mental Health; Ethnically Diverse Groups

**Corresponding author:**
Eleftheria Vaportzis;
Email: E.Vaportzis@bradford.ac.uk

## Abstract

The implementation of electronic health records (EHRs) in mental health contexts has been slow. Reasons for this include concerns from healthcare professionals regarding the collection of sensitive information and the stigma associated with mental health services. Despite the low uptake of EHRs, the benefits include patients feeling empowered and in control of their own treatment. However, ethnically diverse groups often access mental health services through crisis pathways and have been found to disengage with EHRs. The aim of this review was to explore ethnically diverse groups' perceptions of the utility of mental health EHRs and establish perceived barriers and facilitators to access. MEDLINE, CINAHL, EMBASE, Scopus, PsycINFO, PubMed and Web of Science were searched. Included papers mentioned ethnically diverse groups from the 37 listed countries in the Organisation for Economic Co-operation and Development, and included service users, clients or patients accessing EHRs in mental health-care settings. Papers were required to be published between 2009 and 2025. Eight papers met all criteria for inclusion, and three themes emerged: language barriers to EHR access, lack of access to technology and perceived impact of EHRs on access to care. Language barriers to EHR access, no access to technology and stigma were significant issues for ethnically diverse groups due to concerns about who has access to the electronic health data. Benefits of accessing EHRs included easier and efficient access to records. EHRs are critical for modern health systems and further work is required to improve EHRs usage in mental health systems for ethnically diverse groups.

## Impact statement

This systematic review is the first to explore how electronic health records (EHRs) are perceived and used by ethnically diverse groups. The review highlights significant and sensitive issues regarding the ongoing inequalities experienced by ethnically diverse groups in the healthcare system. By identifying the key barriers and facilitators to access EHRs, the findings can inform policymakers and healthcare providers in creating appropriate and inclusive mental health electronic systems. The review is contributing to improving access to and implementation of EHRs and ensuring that mental health services become more responsive to the needs of ethnically diverse groups.

## Introduction

Ethnicity-based health research has increased over the past two decades; however, confusion remains on the preferred terminology, partly due to the lack of a standard definition (Williams et al., 2019; Lu et al., 2022). For the purpose of this systematic review, the term ethnically diverse groups will be adopted. This construct encompasses shared cultural characteristics including religion, language, nationality and dietary practices, which collectively contribute to a sense of group identity (Bonham et al., 2018; Lu et al., 2022). Compared with the overall population, ethnically diverse groups demonstrate poorer health outcomes (Ajayi Sotubo, 2021), and are likely to have adverse experiences from psychological services and experience negative outcomes with their mental healthcare (Barnett et al., 2019). For instance, clinicians' difficulty in delivering high-standard healthcare due to linguistic and cultural barriers (Memon et al., 2016), may prevent ethnically diverse groups from accessing health care services. Thus, ethnically diverse groups enter healthcare services through crisis pathways, which is costly as individuals experience a delay in diagnosis and treatment for their mental illnesses (Bansal et al., 2022). In addition, Smith et al. (2020) highlighted that in mental healthcare, the COVID-19 pandemic increased the mental health inequalities experienced by ethnically diverse groups as access to face-to-face support became difficult with limited alternative routes to care and support. Such findings

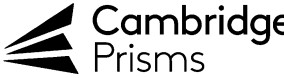

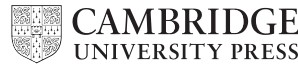

indicate ethnically diverse groups are more likely to experience mental health issues. Consequently, ethnically diverse groups are more likely to report feeling unsafe and may present inadequate knowledge regarding various healthcare settings (Harrison et al., 2020).

As ethnically diverse communities may not frequently use mental health services due to the disparities these individuals face in accessing mental healthcare, such as clinicians poor understanding of the nature of cultural variations in mental health care (Bansal et al., 2022), research has found these groups to disengage with electronic health records (EHRs) (Adams et al., 2024). In modern healthcare, electronic medical records assist professionals in creating treatment plans, and simultaneously share these open medical notes with various other healthcare providers ensuring the continuity of care for these patients (Manca, 2015). EHRs that are not accessible to patients are relevant due to restrictions and clinician practices, which can influence patient engagement and perceptions. This review will focus on patient-accessible EHRs, which are online systems allowing patients to securely view their own medical notes, treatment plans and personal health information from various health contexts such as mental health (Blease et al., 2021). Patients accessing their own EHRs feel empowered and in control of their own treatment (Hägglund et al., 2022). Access to EHRs allow individuals to open their medical notes, this improves patient safety, efficiency and enhances clinical quality (Neves et al., 2020). For instance, Tubaishat (2019) examined the effect of EHRs and highlighted that electronic records can increase patient safety by ensuring data is documented in full correctly and reduce medication errors.

The adoption of EHRs within the context of mental health is slow compared to other health contexts (Kariotis et al., 2022). In some cases, patients with mental health conditions experience limitations which prevent them from accessing their online records (Kariotis et al., 2025). Not all EHRs are fully accessible to patients, particularly within mental healthcare settings as these systems must store sensitive information. Moreover, when records are incomplete and lack medical information, patient's may experience distress as missing information can lead them to relive traumatic events due to their data not being fully documented (Sheikh et al., 2011). For instance, mental health assessments are extremely subjective and in EHRs such observations are stored in clinical note form, which may not provide a complete record of a patient's experience (Garriga et al., 2023). Similarly, Zurynski et al. (2021) highlighted how the uptake of EHRs in mental healthcare settings is limited because structured field within EHRs cannot accommodate patients detailed narrative information. Supporting this, Schwarz et al. (2021) conducted a scoping review and found service users reported adverse experiences with EHRs due to inaccurate notes and disrespectful language, which may negatively affect service user's experiences of their mental health care. Ethnically diverse groups, who already face inequality in mental health services (Bansal et al., 2022), may be particularly affected by incomplete or inaccurate documentation, further contributing to disengagement from healthcare.

It is important to recognise the stigma associated with mental health for ethnically diverse groups as their usage of EHRs is limited (Kariotis et al., 2022). Ethnically diverse groups are less likely to engage with their EHRs due to the stigma experienced by these individuals regarding their mental healthcare (Memon et al., 2016). For instance, research from Memon et al. (2016) revealed ethnically diverse groups are hesitant in discussing their mental illnesses due to the extensive fear of being rejected and receiving negative responses from their communities. Hence, ethnically diverse groups

are reluctant to acknowledge mental health symptoms, and access mental health services (Memon et al., 2016). Blease et al. (2021) aimed to understand the impact on mental health patients in accessing their electronic clinical notes. The results indicated concealing psychotherapy notes in their EHRs could lead to greater patient harm and stigmatisation. This is because EHRs are used to record sensitive data, and clinical information being missed out can be potentially stigmatising for ethnically diverse groups in the context of mental health (Knaak et al., 2017; Kariotis et al., 2022). Sensitive information in EHRs includes personal details such as psychiatric diagnoses, histories of trauma, suicidal thoughts, substance use or psychotherapy notes (Soni et al., 2020). This information can cause concern for patients, making it essential that it is securely protected and safeguarded against manipulation, thereby supporting patients trust and willingness to continue sharing (Keshta and Odeh, 2021). Consequently, these individuals are less likely to engage with EHRs despite the benefits (e.g., having more control of their healthcare). The stigma associated with mental disorders can prevent ethnically diverse groups from engaging with their healthcare due to the concern of social judgement, discrimination and rejection, which could lead individuals to conceal their mental health (Salomon et al., 2010).

Documenting patients' sensitive information regarding their personal behavioural patterns and trauma in EHRs can be considered stigmatising, leaving individuals feeling alienated (Himmelstein et al., 2022). Stigmatising language refers to words or phrases that communicate unintended negative meanings and can strengthen socially constructed power inequities and lead to bias (Shattell, 2009). Himmelstein et al. (2022) conducted a cross-sectional study which aimed to examine the occurrence of stigmatising language in hospital admission notes by assessing 48,651 patient admission notes. They found the prevalence of stigmatising language in EHRs for non-Hispanic Black patients with diabetes compared to non-Hispanic White patients. Stigmatising language may increase these groups' disengagement with healthcare services and EHRs. Instead, making communication inclusive and avoiding labels in health records can disrupt the harmful narratives that allow health disparities to persist (Healy et al., 2022).

Despite stigmatising language in EHRs being an issue for ethnically diverse groups, research from Salomon et al. (2010) highlighted how mental healthcare professionals expressed a low willingness to include confidential and sensitive information in EHRs; a high percentage of these professionals (83%) preferred to limit EHR access to patients. As clinicians in mental health settings would rather not input patients confidential and sensitive information in EHRs, reasons for this include clinician's language use can positively influence clinician-patient relationships, and stigmatising language in EHRs viewed by patients could undermine trust (Desroches et al., 2020; Fernández et al., 2021; Himmelstein et al., 2022). For example, the mistreatment of Black patients when receiving care has resulted in a lack of trust from many ethnically diverse groups in the medical systems (Armstrong et al., 2008). Therefore, with nearly 60% of patients who are offered access and view their EHRs once (Himmelstein et al., 2022), the risk of stigmatising language in EHRs may strengthen the avoidance of taking up electronic records in mental health contexts.

To our knowledge this is the first paper that reviews the experiences of patient accessible EHRs by ethnically diverse groups in mental health settings. Ethnically diverse groups are underrepresented within health and social research (Hussain-Gambles et al., 2004) and primary care services such as mental healthcare (Bansal et al., 2022); thus, these groups experience inequality. This is

concerning as EHRs usability consists of predicting suicide attempts, self-harm, and one's first episode of psychosis, which are all significant mental health concerns (Olfson, 2016; Raket et al., 2020; Irving et al., 2021). Research from Garriga et al. (2023, 2022) explored whether there can be an appropriate identification of determining patients who may be at risk of a mental health crisis, and established the feasibility of EHRs predicting such mental health events which shows the added value of EHRs in mental health clinical practice. For this review, the term EHRs will refer specifically to EHRs that are patient accessible as these systems allow individuals to view their personal mental health records online. Still, it is necessary to include discussion on clinician's use of EHRs and their documentation and practices of the systems and its effect on service user's experiences of accessing their online records. This systematic review is necessary in determining whether there are potential risks to uptake EHRs in mental health settings and what benefit this could have for ethnically diverse service users. The aim of the review was to explore the perceptions of ethnically diverse groups regarding the utility of mental health EHRs and establish the perceived barriers and facilitators to access. The review questions were: How are electronic mental health records perceived and used by different ethnically diverse groups and what are the barriers and facilitators?

## Materials and methods

### Protocol

The review protocol was registered with the PROSPERO international prospective register of systematic reviews (CRD42023494011). The Preferred Reporting Items for Systematic Reviews and Meta-Analyses (PRISMA) guidelines were used to facilitate the development of this systematic review (Page et al., 2021).

### Search strategy

An initial limited search was undertaken to identify articles on the topic. The text words contained in the titles and abstracts of relevant articles were used to develop a full search strategy. The search strategy was piloted on MEDLINE (see Supplementary Material S1 Search Strategy) and then adapted to six other databases: CINAHL, EMBASE, Scopus, PsycINFO, PubMed and Web of Science with the aid of a subject librarian.

### Eligibility criteria

Included papers consisted of adult participants over the age of 18. All genders were included. Qualitative, quantitative and mixed methods studies reporting experiences of participants from ethnically diverse groups using EHRs, in mental healthcare settings were included. This systematic review was interested in ethnically diverse groups living in Western countries. The Country/Organisation for Economic Co-operation and Development (OECD) is an international organisation with 37 countries included. Papers included in the systematic review mentioned ethnically diverse groups from the listed countries on the OECD. In OECD countries, ethnically diverse groups constitute minoritised communities, and make up a smaller proportion of the total population compared with the majority group. Moreover, these ethnic groups make up less than 30% of the overall population and constitute minoritised communities in the OECD member countries, which have similar economic and social profiles. These populations often have distinct

cultural practices, languages and traditions, in comparison to the majority population, and therefore are likely to have a reduced awareness of how to access healthcare. Additionally, eligible studies included service users, clients or patients accessing EHRs in a mental healthcare setting. Moreover, included papers were required to address how mental health EHRs are perceived by different ethnically diverse groups. Only studies focusing on the experience of ethnically diverse groups and usage of EHRs in mental healthcare settings were included allowing real-world experiences to be reflected. Studies which hypothetically asked participants on their perception on EHRs in mental health were excluded. The papers were required to be between 2009 and 2025. The date limits were set based on the American Recovery and Reinvestment Act 2009. This act encouraged the uptake and development of EHRs becoming widely available across the world (Honavar, 2020). Papers were excluded if participants were under the age of 18, did not include ethnically diverse groups from OECD countries or did not discuss mental health EHRs.

### Procedure

Search results were managed using Covidence software (Covidence, 2020) including the removal of duplicate entries, the screening of title and abstracts and of full texts (Figure 1). The first reviewer (AI) screened all the title and abstract papers. Up to 10% of the title and abstract of included papers were reviewed by a second reviewer (EV) and a third reviewer resolved any discrepancies (SP). The full text of selected papers was then assessed in detail against the inclusion and exclusion criteria by reviewer one (AI).

### Data extraction

The full-text papers were examined for eligibility and assessed in detail against the inclusion criteria by AI. The data extracted from each article was completed using a standard template in Excel format and included authors details, title of the article, year of publication, country of publication, study design, participant information which included ethnicity, gender, whether participants were service users, patients or clients with access to EHRs in mental healthcare settings, research aims, type of EHR and main findings (Table 1).

### Quality appraisal

The Caldwell quality assessment tool (Caldwell and Georgina, 2011) and the Critical Skills Appraisal Programme (CASP, 2024) adapted by Surr et al. (2017) were used for quality appraisal of included papers. The CASP quality criteria was used to assess the methodological rigour of the included studies. The quality criteria evaluated whether the research aims and questions were clearly stated, and whether ethical considerations have been addressed in the studies. The CASP also determines the suitability of the study design in relation to the research question, including a clear rationale for its selection. Additional criteria rated the sample size, selection process and description, as well as the reliability and validity of the data collection and data analysis methods. Lastly, the CASP quality criteria evaluates the clarity and relevance of the study's findings and discussion to ensure that conclusions are well supported. This tool allows the standardisation of quality assessment of each paper with various study designs using a series of seven questions (Surr et al., 2017). Papers were provided with an overall quality rating score (i.e., high 11–14; medium 6–10; low

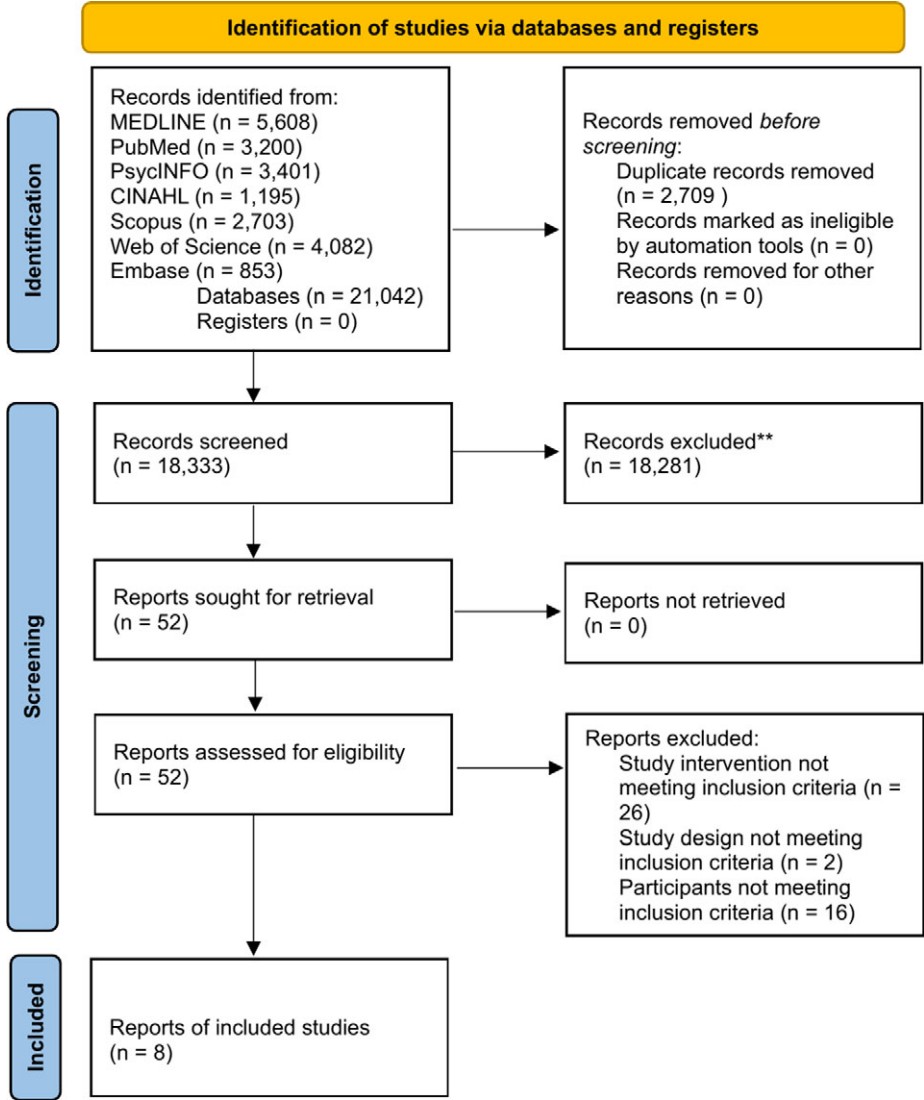

**Figure 1.** This is the PRISMA flow diagram.

score 0–5). High scores indicated the paper met most of the quality criteria, whereas papers with a low score indicated few of the quality criteria was met. All papers were assessed for quality and rated by the author AI (Table 2).

### Analysis

Data analysis was conducted using critical interpretive synthesis (CIS) (Dixon-Woods et al., 2006). CIS provides a systematic, empirical method to combine both quantitative and qualitative research (Bales and Gee, 2012) and involves the process of synthesis of the evidence, allowing the identification of themes to be analysed from each paper. CIS allows reflexivity and critical engagement with the literature (Dixon-Woods et al., 2006). Themes were identified from the language in both quantitative and qualitative papers. Data was analysed iteratively. Findings from the papers in this review were refined to develop an overarching conceptual framework for the barriers, and facilitators on ethnically diverse service users accessing EHRs. Using CIS allows the review question and aims to be understood and determines how shared mental health EHRs are perceived and used by different ethnically diverse groups

and what the barriers and facilitators to access consist of. During the data synthesis process, the results and themes observed were discussed among the three reviewer (AI, SP, and EV).

### Results

In total, eight papers were included in the review, of which six were qualitative (Anthony and Campos-Castillo, 2015; Chang et al., 2018; Gerard et al., 2018; Sadasivaiah et al., 2019; Barreto et al., 2021; Matthews et al., 2022), one was a quantitative study (Bell et al., 2016) and one mixed methods study (Crotty BHM et al., 2015). The qualitative studies included five cross-sectional designs and one semi-structured interview study. The quantitative study was a survey design using Likert scales, open-ended and free text questions. The mixed methods study included monitoring patients and physician's messages on patient portal EHRs. All studies originated from the United States (US) and were published between 2015 and 2022. Most papers included Black/African American, Hispanic, Latino and Asian ethnically diverse groups. Additionally, the types of EHRs mentioned were patient portals in five papers (Crotty

**Table 1.** Data extraction

| Authors | Title | Country | Study design | Participants | Research aims | Type of electronic health record (EHR) | Main findings |
|---|---|---|---|---|---|---|---|
| Anthony and Campos-Castillo (2015) | A looming digital divide? Group differences in the perceived importance of EHRs | United States of America | Qualitative cross-sectional survey | 4,819 respondents 3,760 White 578 Latina/o 481 Black | To examine whether socio-demographic factors, particularly race/ethnicity and gender, as well as health status, health care, and technology, are associated with whether health-care users view electronic access to medical records as important for themselves and for their providers. To understand if groups already at risk of health-care disparities are more or less likely to identify electronic access to health records as important | EHRs | Blacks and Latina/os experiencing psychological distress were more likely than others to perceive EHRs as personally important. Racial and ethnic differences in access to technology. Internet users valued EHRs more than non-users, while those whose providers did not use EHRs – or were unsure – were less likely to see them as important. Racial and ethnic minorities suggest that members of such groups are likely willing to use new information tools that they think can be helpful for their health and healthcare. Patients who may experience poor health care because of their psychological distress may benefit from access to their EHR |
| Barreto et al. (2021) | The role of race, ethnicity, and language in care transitions | United States of America | Qualitative cross-sectional survey | 224 patients 106 males 118 females 68 random sample cross-sections 54 Black (non-Hispanic) 51 Asian (non-Hispanic) 51 Hispanic or Latino | To understand how the transition from hospital to community could be improved with a focus on racial, ethnic, and language differences | Patient Gateway Portal | Hispanic or Latino patients with limited English proficiency (LEP) were less likely to report access to a laptop, desktop, or tablet computer and less likely to access the Patient Gateway portal. Technology barrier may be a barrier to portal access. Despite the availability of the EHR patient portal in Spanish, Hispanic or Latino patients with LEP were less likely to access the Patient Gateway Portal. Black or African American patients are less likely to use the Patient Gateway portal. Lack of access to patient portals, can impact health care engagement |
| Bell et al. (2016) | When doctors share visit notes with patients: a study of patient and doctor perceptions of documentation errors, safety opportunities and the patient–doctor relationship | United States of America | Quantitative (surveys including Likert scales, open-ended and free text questions) | 99 volunteer doctors 4,592 patients White Black/African American other | To investigate whether allowing patients to read their OpenNotes would create a positive doctor-patient relationship | OpenNotes | Non-White patients with fewer years of formal education or had fair or poor self-reported health were more inclined to report feeling better about their doctor after reading their visit notes. Vulnerable populations benefit from reading notes, such as better remembrance of the care plan, feeling in control |
| Chang et al. (2018) | Racial/Ethnic variation in devices used to access patient portals | United States of America | Qualitative cross-sectional analysis | 3,18,700 adult's data 2,18,046 non-Hispanic White 13,172 non-Hispanic Black 25,265 non-Hispanic Asian 14,484 Hispanic 15,978 other 31,755 missing | To examine the racial and ethnic differences in the devices used by patients to access medical records through an online patient portal | Online patient portal on the Group Health Cooperative website. Now known as Kaiser Permanente Washington | Eight key portal functions were identified: secure messaging with providers, requesting medication refills and appointments, and viewing after-visit summaries, medical test results, medical conditions, allergies, and immunisations. Portal use was lowest among Black enrolees, and racial/ethnic minorities were less likely to use active functions but similarly or more likely to use passive ones. Non-English speakers were less likely to use most portal |

(Continued)

**Table 1.** (*Continued*)

| Authors | Title | Country | Study design | Participants | Research aims | Type of electronic health record (EHR) | Main findings |
|---|---|---|---|---|---|---|---|
| | | | | | | | features, especially active ones, indicating language is a barrier to use and not access. Black and Hispanic users were more likely to rely on mobile devices, highlighting disparities in internet access and portal engagement |
| Crotty BHM et al. (2015) | Prevalence and risk profile of unread messages to patients in a patient web portal | United States of America | Mixed methods analysis of 6 years of messages (2005–2010) from physicians to patients on a patient portal | Physicians Patients White Black Asian Hispanic | To assess the prevalence and risk profile of unread messages in a mature patient portal | PatientSite | African American or Hispanic patients in the lowest income group were more likely to have unread messages from physicians on the patient portal named PatientSite (providing patients their electronic data and communicating electronically with their providers) |
| Gerard et al. (2018) | The importance of visit notes on patient portals for engaging less educated or non-White patients: Survey study | United States of America | Qualitative cross-sectional survey | 31,049 patients 11,678 male 19,371 female 1,737 Asian 1,458 Black 765 Hispanic/Latino 23,917 White 1,531 other 1,641 unknown | To understand how vulnerable patients engage with their health information | OpenNotes | Black respondents felt notes were important to understand their health and medical conditions, remember the care plan, understand how the provider is thinking about their medical conditions. Hispanic/Latino and Asian patients reported higher important to notes understanding health and medical conditions, feeling informed about own care, understanding how providers are thinking about medical conditions, remembering care plan, helping make decisions about own care. Study found non-white patients assign high importance to OpenNotes and patients of different races and ethnicities may find transparent notes helpful |
| Matthews et al. (2022) | Acceptability of health information exchange and patient portal use in depression care among underrepresented patients | United States of America | Qualitative semi-structured qualitative interviews | 27 patients 8 male 19 female 3 White 15 Black/African American 1 Non-White Hispanic/Latinx 1 other | To identify patient barriers and facilitators toward the acceptance of Health Information Exchange (HIE) in the context of depression. To examine how HIE affects depression-related care coordination and patient activation | EHR system used is Epic and patient portal is MyChart | Participants had concerns about sharing depression related information due to stigma particularly within Black and African American communities and how this was understood by providers who had access to the portals. Participants expressed quality of care was enhanced when providers were sharing information with one another. The HIE between the providers on the portal made participants feel confident in their providers' recommendations for treatment for their physical and mental health. Participants stated patient HIE allows them to engage with their treatment by tracking their own care and remembering important ongoing health maintenance and symptom |

(*Continued*)

**Table 1.** (*Continued*)

| Authors | Title | Country | Study design | Participants | Research aims | Type of electronic health record (EHR) | Main findings |
|---------|-------|---------|-------------|--------------|---------------|------------------|---------------|
| | | | | | | | reduction, such as the names of medications or their diagnosis. Some participants indicated usability and digital literacy as a barrier to accepting patient portals. Usability concerns are related to the design of patient portals and how easy this is to navigate |
| Sadasivaiah et al. (2019) | Disparities in patient-reported interest in web-based patient portals: Survey at an urban academic safety-net hospital | United States of America | Qualitative cross-sectional study | 16,507 patients 4,362 White 4,652 Hispanic 3,063 African American 3,147 Asian 1,309 other | To examine Web-based portal reported by hospitalised patients, and how interest varied by sociodemographic characteristics, and the barriers among those declining portal interest | Web-based patient portal | Reasons from patients expressing no interest in the web-based portal included: no interest, no ability to use/access computers/internet, does not speak English, physically or mentally unable, does not want to say, security concerns and not useful. Portal interest was the lowest in African American and non-English-speaking participants |

**Table 2.** Quality appraisal

| Article | Clear aims | Ethics | Methodology | Sample | Data collection | Data analysis | Findings | Total (0–14) | Quality |
|---------|-----------|--------|-------------|--------|-----------------|---------------|----------|--------------|---------|
| Anthony and Campos-Castillo (2015) | 1 | 0 | 2 | 2 | 2 | 2 | 2 | 11 | High |
| Barreto et al. (2021) | 2 | 1 | 2 | 1 | 1 | 1 | 2 | 10 | Moderate |
| Bell et al. (2016) | 2 | 1 | 0 | 1 | 1 | 2 | 2 | 9 | Moderate |
| Chang et al. (2018) | 2 | 1 | 2 | 2 | 2 | 2 | 2 | 13 | High |
| Crotty BHM et al. (2015) | 2 | 1 | 0 | 1 | 2 | 2 | 2 | 10 | Moderate |
| Gerard et al. (2018) | 2 | 1 | 2 | 1 | 2 | 2 | 2 | 12 | High |
| Matthews et al. (2022) | 1 | 1 | 2 | 2 | 2 | 2 | 2 | 12 | High |
| Sadasivaiah et al. (2019) | 2 | 1 | 2 | 2 | 2 | 2 | 2 | 13 | High |

*Note:* Quality criteria rating: 0 = No, 1 = Partially, 2 = Yes. 0–5 = Low quality, 6–10 = Moderate quality, 11–14 = High quality.

BHM et al., 2015; Chang et al., 2018; Sadasivaiah et al., 2019; Barreto et al., 2021; Matthews et al., 2022), two papers incorporated Open-Notes, which are EHRs allowing service users to access their clinical notes or visit reports (Bell et al., 2016; Gerard et al., 2018) and one paper included EHRs in general (Anthony and Campos-Castillo, 2015).

### Quality review

Five papers were rated as high quality and three papers as moderate quality based on the Caldwell quality assessment (Caldwell and Georgina, 2011) and the CASP tools (CASP, 2024) adapted by Surr et al. (2017). None of the papers mentioned ethical considerations such as informed consent or confidentiality awareness. Papers from Barreto et al. (2021) and Bell et al. (2016) did not mention the inclusion criteria of their papers explicitly; however, both studies still stated how participants were recruited for the research. Crotty

BHM et al. (2015) did not mention how participants were recruited or the study design.

### Themes

Three themes were drawn from the eight studies included in this review: language barriers to EHR access, lack of access to technology and impact of EHRs on access to care.

### Language barriers to EHR access

Ethnically diverse groups are less likely to have English as their first language (Khan et al., 2021) and consequently experience language barriers when engaging with EHRs. Supporting this notion, Barreto et al. (2021), Chang et al. (2018) and Sadasivaiah et al. (2019) collectively reported a lack of English proficiency was a significant barrier for patients to engage with their own mental healthcare

through EHRs. Chang et al. (2018) found Asians and Hispanics had considerably higher numbers of patients with a primary language other than English, and these patients had lower usage of the EHR patient portal compared to patients with English as a primary language. Furthermore, Chang et al. (2018) found that patients with a primary language other than English were less likely to use the features of patient portal, such as secure messaging, the ability to request appointments, medication refills and viewing the after-visit summaries. Sadasivaiah et al. (2019) also found that non-White, non-English speakers had lower interest in patient portal due to not being able to speak English.

### Lack of access to technology

With health systems using EHRs, the ability to pull and send data from mobile devices allows the requests of diverse patients with various clinical needs to be met (Shaw et al., 2020). Barreto et al. (2021) reported that despite EHR patient portals being available in the Spanish language, Hispanic and Latino patients were still less likely to report access to a laptop, desktop, tablet or computer limiting their access to the patient portals (Barreto et al., 2021). Sadasivaiah et al. (2019) found patients from ethnically diverse groups with no ability or understanding to use computers and access the internet were unlikely to use mental EHRs web-based portals. Chang et al. (2018) also found Black and Hispanic portal users of EHRs were more likely to use mobile devices than a desktop to access the patient portals. Anthony and Campos-Castillo (2015) found racial and ethnic disparities in technology access which contributed to gaps in both access to and opinions about EHRs health information and communication technologies. The study highlighted ethnically diverse groups including Latino and Black groups had less access to technology, and thus, were less likely to benefit from or trust mental EHRs (Anthony and Campos-Castillo, 2015).

### Perceived impact of EHRs on access to care

EHRs allow the successful utilisation of health information exchange with patient's health data being stored, retrieved and updated (Matthews et al., 2022). Supporting this notion, Matthews et al. (2022) illustrated respondents including Black/African American and non-White Hispanic/Latino individuals found elements of patient portal beneficial as access to depression care was easier and more efficient. The messaging and scheduling features on EHR patient portals facilitated quick access to the respondent's providers, and thus, accessibility barriers associated with reaching providers by the phone were overcome (Matthews et al., 2022). The messaging and scheduling features on the EHRs allowed patients to handle their mental health depression care in their own time, instead of relying on service providers or the doctors (Matthews et al., 2022). Furthermore, findings from Matthews et al. (2022) demonstrated service providers receiving detailed information about patient's current mental health data, allowed respondents to feel more confident in service providers recommendations for treatment for depression and various other medical conditions (Matthews et al., 2022). This finding is supported by Gerard et al. (2018) as they found EHRs allowed less educated and non-White patients, including Asian, Black and Hispanic/Latino ethnically diverse groups to understand their health more. More specifically, Black respondents Hispanic/Latino and Asian patients found EHRs allowed an understanding of health and medical conditions, feeling informed about their own care, recognising how providers are thinking about medical conditions, remembering care plan and

helping make decisions regarding their own care (Gerard et al., 2018). In line with this finding, Bell et al. (2016) reported African American patients benefited from reading their notes as they better remembered their care plan, became more in control and were taking medications better as prescribed. Bell et al. (2016) also illustrated how patients from ethnically diverse backgrounds have a distrust for the healthcare system. However, the research demonstrated sharing notes on EHRs could result in the patient and providers views being similar and may influence patient's in perceiving their doctor positively (Bell et al., 2016). It is also important to recognise the findings from Anthony and Campos-Castillo (2015), which revealed Black and Latino patients with psychological distress were most likely to report that the EHRs were important for themselves, even after controlling for respondents' socio-economic status, health status, health care context and disposition toward health information. Moreover, compared to White patients, Black and Latino patients with psychological distress who were more likely to experience healthcare disparities perceived EHRs as being very important for themselves (Anthony and Campos-Castillo, 2015). Supporting this notion, Anthony and Campos-Castillo (2015) highlighted that respondents' specifically Black and Latino patients' level of psychological distress was positively associated with perceiving EHRs as personally important. Furthermore, the paper revealed the perceived importance of EHRs for ethnically diverse groups and highlighted their willingness to use new information tools such as EHRs as they believed they could be helpful for their personal healthcare (Anthony and Campos-Castillo, 2015).

Alternatively, ethnically diverse groups do present concerns of mental health-related patient portals. For instance, Matthews et al. (2022) found exchanging information related to depression within EHRs was linked to the stigma associated with the mental illness (Matthews et al., 2022). Thus, there was a concern among ethnically diverse respondents with how their mental health-related information was used or perceived by other healthcare providers when accessing the EHRs (Matthews et al., 2022). These attitudes were more prevalent among the racially and ethnically diverse patients as respondents described fear of stigma, labelling or unauthorised disclosure of their mental illness due to health information being stored and retrieved in patient EHR portals by service providers (Matthews et al., 2022). Hence, the stigma associated with mental health illness, specifically depression, inhibited acceptance of electronic communication and information sharing in EHRs by ethnically diverse group respondents (Matthews et al., 2022).

### Discussion

This systematic review explored ethnically diverse groups perceptions of the utility of mental health EHRs and reports perceived barriers and facilitators to access. Research within this field is limited, with the systematic review identifying eight relevant studies, all originating from the US. This demonstrates a lack of research on ethnically diverse groups outside the US. Limited evidence on ethnically diverse groups outside the US may reflect methodological and ethical constraints rather than low EHR adoption. For instance, asking about ethnicity within survey research is not common practice, and collecting these data vary widely (Mauro et al., 2022).

With the systematic review identifying studies all originating from the US, some have suggested that EHR implementation outside the US is slower (Kariotis et al., 2022). However, this may not be the case, as countries in northern Europe and Estonia

provide patient's access to mental health records (Bärkås et al., 2021). In the US, the Health Information Technology for Economic and Clinical Health (HITECH) Act of 2009 was developed to increase EHR adoption (Adler-Milstein and Jha, 2017). For instance, in the US, EHR use has increased in general medical outpatient (85.9% in 2017) and inpatient (96% in 2017) settings (Spivak et al., 2021). Still, adoption may be slower among mental health facilities with less than half of all psychiatric hospitals reporting certified EHR use in 2016 (Hu et al., 2020).

Nevertheless, within Europe, patients have some access to their mental health records through national EHR systems (Bärkås et al., 2021). For example, in Sweden, patients in mental health care accessed psychiatric notes, however, the research illustrated challenges such as errors, omissions and stigmatising language in records (Bärkås et al., 2023). Similarly, Honey et al. (2024) explored the experiences of people using mental health services in Australia and found participants felt uncertain about the information held and accessed by different organisations. Participants recommended that their mental health records be collected and maintained transparently, with information used only to support their care (Honey et al., 2024). These examples highlight that EHR use in mental health settings varies internationally, and caution is needed when generalising US-based findings. Moreover, the studies by Bärkås et al. (2023) and Honey et al. (2024) did not explicitly include ethnically diverse groups, illustrating the need for research that explores these individuals' perceptions of mental health EHRs. Ethnically diverse groups in the US differ culturally from those in other countries, and health systems vary in terms of EHR implementation and patient access.

The review found language barriers to EHR access being a barrier to accessing and utilising EHRs. This echoed in wider research. For instance, Sentell et al. (2007) aimed to understand language barriers in mental health care and found limited English proficiency was associated with lower use of mental health care. More specifically, non-English speaking Asians and non-English-speaking Latinos were less likely to receive services compared to Asian and Latinos who speak English (Sentell et al., 2007). This reflects the finding from this review as Asians and Hispanics were more likely to experience language barriers making it difficult for these ethnically diverse groups to engage with EHRs. Such barriers indicate the need to provide translated materials and interpreter support to ensure that ethnically diverse groups can understand their medical information, which is essential for successful utilisation of EHRs (Barreto et al., 2021).

Moreover, it is important to recognise that English is the native language of only 7.3% of the world's population and less than 20% can speak the language (Bahji et al., 2023). Therefore, while language proficiency is often cited as a barrier to accessing EHRs in English speaking countries, most EHRs globally are written in local languages (Barreto et al., 2021; Bärkås et al., 2023). Linguistic barriers, therefore, affect ethnically diverse groups differently depending on the national context.

The lack of access to technology was found to be a barrier in accessing EHRs, as Black, Asian, Hispanic and Latino ethnically diverse groups were less likely to possess a device to access EHRs. Technological devices are being increasingly used in the healthcare sector, as mobile devices allow the continuous updating of patient information and improve the communication between the service providers and service users (Heponiemi et al., 2021). For instance, Ennis et al. (2012) explored the extent to which mental health service users have access to and skills in using various technologies. The research illustrated Black and ethnically diverse groups were more likely to access computers outside of their own homes than White individuals (Ennis et al., 2012). This suggests ethnically diverse groups should be given access to the technology if healthcare services expect individuals to access EHRs, to allow successful engagement with the services (Ennis et al., 2012). This reflects the findings from this review as Black, Hispanic and Latino ethnically diverse groups had less access to laptop, desktop, tablet or computer making it more difficult for these groups to utilise EHRs. Lack of access to the appropriate technology may be a significant barrier which contributes to the digital divide in these ethnically diverse groups attempting to access EHR portals.

It is important to recognise that barriers to lower technology access and use may be related to demographic characteristics including household income (Arcury et al., 2017). Crotty BHM et al. (2015) suggested African Americans, Hispanics and individuals with household incomes below $25,000 had more unread messages on the patient portals despite having access to a computer and the internet. Further research is required to understand the barriers these groups experience and help overcome them allowing beneficial engagement with EHR patient portals.

Stigma was found to be a perceived barrier for service users to consent to health information exchange through EHR patient portals (Matthews et al., 2022). Stigma around mental illness is prevalent within Asian, Black and African American communities (Eylem et al., 2020), as this can prevent these groups from engaging with their healthcare due to the worry of social judgement, discrimination and rejection which could lead to individuals to conceal their mental health difficulties (Clement et al., 2015). Thus, explaining the reviews findings of ethnically diverse groups concerns of how their mental health related information was used or perceived by other healthcare providers when accessing the EHRs (Matthews et al., 2022). Supporting this notion, Himmelstein et al. (2022) conducted a cross-sectional study and found that non-Hispanic Black patients notes included words such as "unwilling," "refused," "noncompliance" and "refuses." Hence, stigmatising language in EHRs is often used to describe non-Hispanic Black patients which can alienate these patients and may result in adverse and inequitable health outcomes for patients (Himmelstein et al., 2022; Bilotta et al., 2024). These results illustrate negative words and labels associated with ethnically diverse groups can affect their acceptability of EHRs, and thus, safeguarding individuals against stigma could allow the expansion of EHRs in mental health settings.

Despite the barriers, this review found EHRs allowed mental health care for depression to become more efficient. Ethnically diverse groups became more informed about their own care, recognised how providers are thinking about medical conditions, remember care plans and make decisions regarding their own care due to EHR access (Gerard et al., 2018; Matthews et al., 2022). Supporting this notion is research from Manca (2015), which suggested EHRs improve quality of care, patient outcome and safety and improve access to patient data as communication between healthcare providers and patients is enhanced. Moreover, this review found that Black and Latino patients with psychological distress who have experienced healthcare inequalities perceived EHRs as being very important (Anthony and Campos-Castillo, 2015). These findings are in line with research from Hägglund et al. (2022), which suggested EHR systems allow patients to feel empowered and in control of their own treatment.

### Limitations and strengths

Many studies were identified in this systematic review and for this reason a fully blind review with two reviewers screening the title and

abstract was not feasible. This is a limitation of the study due to potential selection bias. However, reviewer one (AI) completed the full abstract and title screening. Up to 10% of the title and abstract of included papers were reviewed by a second reviewer (EV). The third reviewer resolved any discrepancies (SP) helped ensure consistency and reliability in the screening process. The involvement of multiple reviewers and the resolution of discrepancies strengthened the rigour and credibility of the review. A further limitation is that ethnicity is not routinely collected in survey or clinical data due to ethical considerations. As discussed earlier, this may partly explain the lack of evidence from outside of the US, reflecting methodological and ethical constraints rather than differences in EHR adoption. Also, only including OECD countries is another limitation, as the review excludes information from low- and middle-income settings, where experiences with EHRs may differ significantly. This may reduce the generalisability of the findings. Moreover, this review included articles published in English and may have excluded non-English language studies that may have contributed to understanding EHR usage by ethnically diverse groups in mental health. Despite the limitations, utilising a rigorous systematic review methodology and including quantitative, qualitative and mixed-method studies are major strengths. Similarly, conducting an in-depth literature search across multiple databases reduces the risk of missing relevant studies. The systematic review also applies clear eligibility criteria, which enhances transparency and reproducibility.

### Implications of the results for practice, policy and future research

The findings from this systematic review highlight implications for practice, policy and give directions for future research in improving ethnically diverse groups utilisation of mental health EHRs. Mental healthcare providers should consider ways to overcome language barriers, such as offering translated EHR systems. For instance, future research could determine whether ethnically diverse service users in mental health would benefit from translating systems on EHRs which could identify difficult terms, replace them with alternative synonyms and generate explanatory texts within the EHRs. Ethnically diverse groups lack of access to technology leading to limited understanding of the patient portals demonstrates the need of digital literacy programs empowering individuals to utilise the EHR systems effectively. Mental healthcare providers should focus on reducing technology inequalities by providing appropriate devices and increasing the availability of user-friendly EHR platforms for ethnically diverse groups. These challenges are not limited to mental health settings. For instance, language barriers, digital literacy gaps and technology inequalities may influence access and engagement with EHRs across different clinical contexts, such as primary care, chronic disease management and other health services. Policies should highlight privacy protections to relieve the concerns about stigma and the potential misuse of mental health information. Again, issues around confidentiality and trust in data security are vital for patient engagement with EHRs in general health care settings and not just in mental health. Future research should investigate the long-term impact of EHRs on mental health outcomes among ethnically diverse groups and investigate strategies to lessen stigma associated with mental EHR usage. Future research should investigate how factors such as socioeconomic status, education level and digital literacy may affect ethnically diverse groups engagement with mental health EHRs, particularly in diverse healthcare systems outside the US. Addressing these gaps

can improve access to and benefits from EHRs for all ethnically diverse groups in mental healthcare.

### Conclusion

EHRs research in the context of mental health is limited. The review is the first to explore how electronic mental health records are perceived and used by different ethnically diverse groups and begin to determine the barriers and facilitators to access. The current systematic review identified and included eight papers. Language barriers to EHR access were identified as a barrier for ethnically diverse groups understanding their clinical information on EHRs. Technology inequalities prevent these groups from accessing EHRs. Mental healthcare providers need to consider providing ethnically diverse groups with translated EHRs and access to the appropriate technology to overcome the barriers and encourage successful utilisation of EHRs. Stigma was also found to be a barrier among ethnically diverse groups regarding how they may be perceived due to their EHR notes. The review illustrated that ethnically diverse groups became more informed about their own care, recognised how providers are thinking about medical conditions, remembered care plans and made decisions regarding their own care due to mental health EHR access. Further work is required to reduce the barriers in utilising EHRs in mental health settings and ensure ethnically diverse groups can safely access their electronic notes.

**Open peer review.** To view the open peer review materials for this article, please visit http://doi.org/10.1017/gmh.2025.10063.

**Supplementary material.** The supplementary material for this article can be found at http://doi.org/10.1017/gmh.2025.10063.

**Data availability statement.** The authors confirm that the data supporting the findings of this study are available within the article and its supplementary materials.

**Author contribution.** AI: Methodology, formal analysis (Spivak et al., 2021; Kariotis et al., 2022), investigation, writing – original draft, project administration. SP: Conceptualisation, investigation, methodology, writing – reviewing and editing, supervision. EV: Conceptualisation, investigation, methodology, writing – reviewing and editing, supervision, funding acquisition.

**Financial support.** The research is a UKRI Brad-ATTAIN funded project.

**Competing interests.** The authors declare no conflict of interest.

**Ethics statement.** Ethical approval was not required for the systematic review of available and accessible literature. The protocol was registered with PROSPERO (CRD42023494011).

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
