## [Reviewer Report]

Dear Author,

This article is very informative, as it provides a comprehensive review of the literature regarding Electronic Health Record (EHR) adoption from the patient perspective. analysis of telehealth utilization and motivations for telehealth use by patient. Readers will benefit from learning about patient perceived barriers to accessing and using EHR when receiving mental health care.

Overall, I don’t have major suggestions.

I do, however, have one minor suggestion regarding terminology. Lately, the word “minority” has been deemed offensive across groups. While you do provide a definition from Reid and Mabhala (2021), I don’t necessarily agree with the authors' definition as it groups several patient demographics. Research has shown that disparities differ across demographic characteristics. For example, racially diverse patients may encounter different access barriers than patients who come from a different country (nationality). Moreover, in the US, for example, there are cities and states that “minority” groups are actually the majority, so this term can be misleading at times.

I strongly discourage the use of the term “minority ethnic groups”. Also, your literature review focuses mostly on race, ethnicity, and language. Alternatively, I’d suggest looking within the demographic characteristics that came up during your literature search and come up with a different term to name your target population. You can use, for example, “diverse groups”, “diverse patient populations”, “racially and ethnically diverse patients”, etc.

---

## [Reviewer Report]

A competent systematic review but one with limited novel content of which adds anything of value to the extant literature.

---

## [Reviewer Report]

The paper presents a systematic literature review of the perceptions of electronic health records among minority ethnic groups in mental health care. The review is well-written and clearly presented. The topic is timely and important given the increased implementation of patients' online record access and electronic health records. There are however some room for improvement, especially when it comes to the contextual and global perspectives of the analysis, and the interpretation and implications of the findings.

I will now provide more detailed comments based on each section of the paper.

INTRODUCTION

1. The introduction presents the topic of minority groups and mental healthcare well. The description of EHRs in general and patients’ online access to EHRs could however be improved. In the introduction, it is not entirely clear if the review will focus exclusively on EHRs that are accessible to patients, or on all EHRs used in mental healthcare. The distinction is important in this study, as patients with experience of accessing their own records may have different perceptions of EHRs than patients who have not. On the other hand, as a large focus is on patient engagement with their EHR and the barriers to this, are EHRs that are not accessible to patients of relevance at all to the study? This should be more clearly described in the introduction.

2. On line 74, Yan et al. is used as a reference to support that minority ethnic groups disengage with EHRs. However, the study focuses on healthcare professionals and indicate that they do not engage as much with the EHR when treating patients from minority group. Please check the reference.

3. Boonstra et al is used as a reference to argue that “as EHRs allow individuals to open their medical notes” they improve care in different ways, however the study does not at address patients online record access at all – it only looks at how EHRs are used by clinicians and how the structured documentation and administration required affects clinicians’ autonomy and dialogue with patients. Please either find a better reference or rephrase.

4. The first paragraph on page 4, lines 84-99 needs to be rephrased for clarity. The sentence starting on line 85 is difficult to understand – how is the requirement to store sensitive information related to incomplete records – and how does incomplete records cause distress?

5. Sensitive information and stigmatization is often group together throughout the manuscript. I would like to see a more in-depth description of what may constitute stigmatizing language in an EHR vs. what could be considered sensitive information. Information that is sensitive may be described in a respectful way, and would then not be considered stigmatizing. These are important distinctions as they may relate to different barriers addressed in this study; barriers to patients’ own engagement with EHRs, and patients’ concerns about the implementation and use of EHRs that enable sharing of their records with a wider audience of healthcare professionals – which might become a barrier for seeking mental healthcare in the first place.

METHODS

6. The reason for focusing on OECD countries is not clearly argued. Why only these countries?

7. I would have expected a fully blinded review with at least 2 reviewers screening each abstract, but upon seeing the number of papers you identified I understand why you did not do this. However, I believe this should be discussed in the manuscript, as a potential weakness of the study.

8. Please provide more detail under the heading ‘Analysis’. How were e.g. qualitative and quantitative findings combined?

9. Was it a requirement that all included studies focused on actual experience of using EHRs? Or could studies that more hypothetically asked patients about their opinions of EHRs also be included?

RESULTS

10. Line 245-246 states that 2 papers did not “mention the inclusion criteria of their papers”. Does this refer to inclusion criteria for the participants in their studies? Although technically correct (they do not use the term ‘inclusion criteria’) both studies describe how respondents were recruited (either in the referred to study, or in the case of Bell et al – in the referenced paper describing the original study). I wonder if this should be revisited?

11. The first theme “Limited English proficiency as a barrier” could perhaps be renamed to facilitate transferability? While true in the US and other English-speaking countries, in other countries not speaking the majority language (whether Finnish, German or Spanish) would rather be the barrier. Renaming the theme to e.g. ‘Language barriers to record access’ could be a way to make the results more transferrable to other contexts (while still true).

DISCUSSION

12. Only 8 out of over 18 000 identified studies were included, and all were from the US. This needs to be discussed more in-depth. You argue that EHRs are not widely implemented in mental healthcare which may be true, but not really in e.g. the Nordic countries and Estonia where patients have substantial access to their MH records (see e.g.

Bärkås, A.; Scandurra, I.; Rexhepi, H.; Blease, C.; Cajander, Å.; Hägglund, M. Patients’ Access to Their Psychiatric Notes: Current Policies and Practices in Sweden. Int. J. Environ. Res. Public Health 2021, 18, 9140. https://doi.org/10.3390/ijerph18179140

Bärkås A, Kharko A, Blease C, Cajander Å, Johansen Fagerlund A, Huvila I, Johansen MA, Kane B, Kujala S, Moll J, Rexhepi H, Scandurra I, Wang B, Hägglund M. Errors, Omissions, and Offenses in the Health Record of Mental Health Care Patients: Results from a Nationwide Survey in Sweden. J Med Internet Res 2023;25:e47841. doi: 10.2196/47841

13. You argue in the discussion that implementation of EHRs in countries outside the US is slower. I would argue that this is not the case, looking at e.g. northern Europe and Australia, patients have extensive online record access. However, and this is important, especially in northern Europe – asking about ethnicity and race is not common practice in survey research. In fact, special caution is advised by ethical committees when asking such questions, and they are often avoided. Instead, other types of questions are asked, such as country of birth, language proficiency etc to determine whether an individual belongs to a minority group. Given your extensive search terms, studies should still have been found – and we can assume that they do not exist, however to say that they do not exist because EHRs are not implemented is simply not true.

14. In addition to the point above, while EHRs are very widely implemented across Europe (and have been in use for many years), patients’ online access to them is a more recent phenomena in many countries (excluding northern Europe where many countries gave patients online access around 2010 – see ref below). Yet, when you don’t distinguish between EHRs in general and patients’ online record access to EHRs the arguments sometimes becomes incorrect.

Moll J, Scandurra I, Bärkås A, Blease C, Hägglund M, Hörhammer I, Kane B, Kristiansen E, Ross P, Åhlfeldt RM, Klein GO

Sociotechnical Cross-Country Analysis of Contextual Factors That Impact Patients’ Access to Electronic Health Records in 4 European Countries: Framework Evaluation Study

J Med Internet Res 2024;26:e55752

doi: 10.2196/55752

15. Again, while English language may be a barrier in English speaking countries – please highlight here that most EHRs globally will not be written in English.

16. Please expand the section ‘Limitations and Strenghts’. Add e.g. a discussion about your search strategy, the fact that only one reviewer screened most abstracts, and how limitations in reporting ethnicity in non-US contexts could limit the identifiability of studies. The limitation to OECD countries could also be described as a weakness.

17. Please consider whether some of your results can transfer to other clinical contexts than mental health. Language barriers and lack of technology might e.g. be equally important barriers for minority groups in other clinical settings.

CONCLUSION

18. Make sure that the conclusion is updated to reflect any changes made to e.g. language barriers.

MINOR REVISIONS

1. The first sentence of the introduction has an “in” too many (line 57).

2. Line 178: remove ‘,’ after ‘mental health’.

3. The reference Honavar 2020 (line 181) is missing from the reference list.

4. Line 183: Should read ‘did not discuss mental health EHRs.’?

5. Line 215: Assuming AI is the first author and not Artificial Intelligence, I would clarify this here (considering the increasing use of AI in scientific analysis).

6. Line 289: ‘services’ should be ‘service’.

7. Line 292: ‘in-service’ should be ‘in service’.

---

## [Reviewer Report]

Thank you for carefully considering and responding to my earlier comments. I believe your thoughtful responses and edits have strengthened the manuscript, and from my perspective, I have no further comments.